# Design Space for Voice-Based Professional Reporting

**Jaakko Hakulinen** [1,*][ID]**, Tuuli Keskinen** [1]**, Markku Turunen** [1] **and Sanni Siltanen** [2]

[1] Faculty of Information Technology and Communication Sciences, Tampere University, P.O. Box 1001, 33014 Tampere, Finland; tuuli.keskinen@tuni.fi (T.K.); markku.turunen@tuni.fi (M.T.)

[2] KONE Corporation, KONE Technology and Innovation, Myllykatu 3, 05800 Hyvinkää, Finland; sanni.siltanen@kone.com

[*] Correspondence: jaakko.hakulinen@tuni.fi

**Abstract:** Speech technology has matured so that voice-based reporting utilizing speech-to-text can be applied in various domains. Speech has two major benefits: it enables efficient reporting and speech input improves the quality of the reports since reporting can be done as a part of the workflow without delays between work and reporting. However, designing reporting voice user interfaces (VUIs) for professional use is challenging, as there are numerous aspects from technology to organization and language that need to be considered. Based on our experience in developing professional reporting VUIs with different stakeholders representing both commercial and public sector, we define a design space for voice-based reporting systems. The design space consists of 28 dimensions grouped into five categories: Language Processing, Structure of Reporting, Technical Limitations in the Work Domain, Interaction Related Aspects in the Work Domain, and Organization. We illustrate the design space by discussing four voice-based reporting systems, designed and implemented by us, and describing a design process that utilizes it. The design space enables designers to identify critical aspects of professional reporting VUIs and optimize those for their target domain. The design space can be used as a practical tool especially by designers with limited experience on speech technologies.

**Keywords:** design space; voice user interfaces; professional reporting; voice-based reporting; speech recognition; speech-to-text

## 1. Introduction

Speech is an efficient way for humans to communicate information and it is used to dictate different types of reports, for instance. A prime example is the medical field where doctors have dictated medical reports for decades already. Traditionally, these reports have been manually transcribed but speech recognition technology has also been applied. Focused specialties with specific language (like radiology) were the first areas where speech technology could be successfully applied to reporting. Speech technologies, and speech recognition, in particular, have evolved constantly for decades so that, today, voice-based reporting with speech recognition is possible in various fields, including areas such as industrial maintenance work, where reporting can happen in the field, during actual maintenance work.

User-centered design builds on understanding people, activities, contexts, and technologies [1]. Of these, the design of voice-based reporting must consider all, but the activities and context must be analyzed on a particularly wide scale and thorough understanding of the work where reporting is to be applied. The target organization must be involved and understood, and the language used in the work domain is important as well. As reporting is part of the users' work, understanding their work overall and the related circumstances is critical. Voice-based reporting can be particularly efficient when it is integrated to the work so that the reporting happens at the right time and the system supports both the work and the reporting.

To achieve this thorough understanding, the involvement of several people is usually required. Many designers are not familiar with speech technologies either [2], so we believe that the design space presented in this article will help to comprehend how various elements link to the design of voice-based professional reporting. While methodologies to collect the data can be the same [3], often the level of detail required can be high due to the need to understand both the work and the language used as well as their implications to technology. In different work domains, this can mean very different reporting systems. In some cases, such as a nurse or a doctor having a patient's appointment, traditional dictation type of reporting, where each report is done in a single session, may be efficient, while other domains, or tasks even, call for highly interactive solutions where reporting is done as a dialogue between the user and the system during the main task. For example, a yearly maintenance visit to check equipment may take hours and create a lot of information that needs to be saved, i.e., reported.

There are various elements in voice-based reporting interfaces that differ from interfaces based on other modalities (mainly graphical user interfaces). Together, the need to understand both the work practices and voice as an interaction modality create a complex field for interface designers to analyze. This is further complicated by the fact that reporting is inherently linked to the information systems and practices of the organization where the work takes place.

We have developed voice-based reporting systems to various domains over the years. Based on our analysis building on this experience, we introduce a design space that consists of 28 dimensions to analyze. By understanding these aspects, designers can consider the multiple factors that influence the design of voice-based reporting systems. We have grouped the dimensions into five categories: Language Processing, Structure of Reporting, Technical Limitations in the Work Domain, Interaction Related Aspects in the Work Domain, and Organization. An eight-step design process is also given to enable focusing on a subset of elements at a time. The design space enables designers to identify critical aspects of professional reporting VUIs and optimize them for each target domain. The design space can be used as a practical tool especially by designers with limited experience on speech technologies.

The paper is organized as follows. Related work on voice-based reporting and design spaces are covered first. Then, we describe four voice-based reporting systems we have developed. These example cases are the material we built the design space on and they are also used later in the paper to illustrate the different aspects of the design space. After the case descriptions, we introduce the design space, listing each of the dimensions under the five categories. This is followed by a process model for designing voice-based reporting systems. Discussion and conclusions form the last section of the paper.

## 2. Related Work

### 2.1. Voice-Based Reporting

Speech technologies have developed continuously for decades, and today, a wider adoption of speech technology has taken place in consumer products. Personal assistants like Apple Siri, Amazon Alexa, and Microsoft Cortana have become popular, especially among native English speakers. Voice search is also a commonly used feature, as speech is an efficient way to enter information, especially in mobile context. Automotive area has also featured speech interfaces for several years and recent solutions have increasingly advanced interaction capabilities and better understanding of free-form voice input.

Speech has been used for reporting or documentation for a long time. In this area, the efficacy of speech in entering information has been the key reason for voice-based reporting. In the medical field, doctors have dictated their reports for a long time—first as audio records transcribed manually and since the 1980s [4], also with speech recognition in various specialties, locations [5], and technologies from multiple companies [6]. The most successful applications have been in areas where language and terminology are very specific so that optimized language models could be created. Radiology has been a prime

example of this [7], but speech recognition is commonly utilized across the medical field including doctors of different specialties [8], nurses [9,10], and dentists [11]. Its benefits for secretaries appear less obvious [12]. Outside the medical field, voice-based reporting has been utilized in legal work [13], and some of the earliest applications can be found in space and aviation fields [14].

Over the decades, the use of speech-based reporting has evolved [15] and improvements in speech technology have reduced the number of recognition errors. Errors are a critical factor in healthcare where the accuracy of the reports is key. Early systems also limited the speaking style, making reporting inefficient [16], but today the situation is better. Today's technology is good enough so that many, but not all, users are willing to utilize speech-based reporting [4]. Understanding and managing user acceptance [15,17] is always important.

Currently, the biggest benefits of speech-based reporting can be found in efficiency and turnaround time [18–22]; with speech technology, the reports are available quickly while manual transcription loop is often slow with queues of several days, even. However, especially for non-native speakers, the actual reporting can still take more time [23]. The downsides are related to the time the error correction takes. To users, the errors tend to appear as an inconsistently behaving system [24].

Other benefits identified for voice-based reporting include the quality and consistency of reports, cost savings, and automatic processing of results that can add semantic information to the reports, structure them automatically, and so on [23,25,26]. The benefits have been studied most extensively in the medical field, but more evidence is still needed [27]. In the medical field, structured reporting and dictation have been contrasted in the past [24], but the combination of the two is considered the goal [28–30]. Natural language understanding has been utilized in medical reporting systems both to structure the reports and to detect important concepts automatically [31–33].

In the industrial area, the usage of speech technology is not yet so widespread. While systems have been in real use for years [34], speech is mostly considered as a way to interact with information systems, e.g., with wearable systems. In industry, speech is often considered particularly useful when workers need to access information in the field, possibly in hands-and-eyes-busy situations. In industrial settings, noisy conditions are one aspect that needs to be considered in relation to speech input [35].

### 2.2. Design Spaces

In the literature, we can find two ways to consider design spaces. The first defines what can be designed and provides a taxonomy type of categorization, often consisting of orthogonal dimensions. The second is a space formed by design artefacts like documents and prototypes created in a particular design process. The latter approach is sometimes considered to include all potential artefacts that could be created [36], moving it closer to the first type. Dimension-based, taxonomy type design spaces relevant to current topics include design space for multimodal interfaces by Nigay and Coutaz [37]. Design space literature has considered different approaches to design. Utilizing design spaces for interaction design is possible [38] and design space analysis has been one of the tools in the design of interactive systems [39,40]. Bowen and Dittmar [41] consider user-centered design, interaction design research, and engineering approaches to human-computer interaction.

Benefits of utilizing design spaces are in the way they can support designers by helping identify relevant parameters of design. Dove, Hansen, and Halskov [41] mention that design spaces can also

- increase the awareness of constraints with certain design choices
- help understand how design activities and (chosen) constraints have led to particular design
- help challenge the (chosen) constraints and realize opportunities.

Braun et al. [42] state that a design space can aid for research by pointing out interesting opportunities and it can establish common ground for designers. Card and Mackinlay state

the design space can help "to understand the differences among designs and to suggest new possibilities" [43]. Design space can create basis for discussing challenges and issues [44]. Haeuslschmid, Pfleging, and Alt [45] mention that design space can "support designers' creativity" and allows exploration of multiple dimensions.

Design involves people with different roles and backgrounds [36]. Bowen and Dittmar [39] state that "in order to achieve quality design, different design options have to be explored and compared and choice should be informed by the viewpoint and expertise of the relevant sub teams." The need for different types of experts is particularly true for speech-based reporting where user interface design and software engineering must work with speech technology and understand the workflows and communication needs within organizations. The concept of design spaces is one tool to help the communication within the design teams.

Various design spaces have been presented in literature over the years. Methods of defining and applying design spaces mentioned in publications include literature reviews and expert interviews [42]. Still, most of the work in defining a design space consists of analysis work by the authors. This is natural as design spaces are on metalevel of design and therefore defining a process to specify a design space is quite useless except for the often-repeated design tasks with specific areas. One systematic approach was presented by MacLean et al. [46], who presented design space analysis method built on question, options, and criteria. Design spaces for interactive systems are rather complex and a generic model is hard to define. Design spaces themselves, however, can be useful. We see the most valuable use of design spaces, when a designer starts to work with a topic not yet familiar to them.

Our work primarily represents the design space approach of identifying dimensions that affect the design of voice-based reporting applications for professional use. We see that the benefit of our work is in helping designers understand the possibilities and limitations when designing speech-based reporting systems. This work guides in identifying the related information that affects the design. Our collaboration with industrial companies and public organizations indicates that, now, as speech technology has reached maturity, it can be applied to different domains without excessive work. For example, there is not necessarily a need to create domain-specific language models from scratch, and when the need arises, tools to do so exist and processes are well supported. Still, speech-based applications include many aspects not encountered in daily development of applications with commonly utilized interaction modalities and techniques. This work should therefore support those who are tasked with development of such systems, making the design process more efficient as the design space can help ask the right questions early in the process and understand the restrictions and possibilities when they can still affect major decisions.

## 3. Materials and Methods

The design space we present in this article is built on learnings from the development and evaluations of four voice-based reporting applications. We developed them in separate projects with different collaborators and designed them for different users and organizations. We first identified these learnings, then categorized them in an iterative process as a group, and finally we specified a process to help the utilization of the resulting design space. Here, we will describe the example systems and explain the process of forming the design space.

### 3.1. Voice-Based Reporting Systems Used as Material

We have developed numerous voice-based reporting systems over the years. In the following, we present four such systems to illustrate the types and requirements of systems alike. We describe the systems as they were planned with people working on each domain. All the solutions were developed in research projects together with representatives of the end users. While none of them has been used in real work outside of the evaluations,

the development included identification of requirements and challenges in the areas. The examples are used to illustrate the design space presented later in the paper.

### 3.1.1. Lawyer Dictation

The lawyer dictation system [47] was developed together with lawyers from a company called Mobiter Dicta to be a tool for dictating letters and legal documents. The application was designed to support dictation of documents in fragments, i.e., a user could dictate small parts of the document at time and continue when the next opportunity arose. The interface running on Symbian mobile phones visualized the detected voice, allowed the user to jump inside and between recordings using physical buttons in the mobile device and add, delete, and replace the recorded audio samples. Speech recognition took place in a server where a large-vocabulary speech recognizer processed the audio. There was noteworthy delay (minutes for longer dictations) in processing the speech, and the users had to explicitly upload the audio to the server. The results could then be reviewed in the mobile device and the user could still switch between the audio files and the text format and do further recordings and edit the recognized text, e.g., by utilizing n-best lists provided by the recognizer. The final documents are the primary contribution of the users. Their documents were sent to be finalized with traditional word processing tools when there was time for such editing. This reduced the need to be able to carefully proofread the results with the application itself.

### 3.1.2. Nurse Dictation

Together with nursing science experts from the University of Turku, we developed a dictation system for nurses [48] and evaluated the prototype with Turku University Hospital. While doctors have long used dictation, nurses have not used the technology as extensively. There are many different areas where nurses work, and their needs vary greatly. We developed the solution to nurses who meet and operate patients on one-to-one basis and need to report the work done after each patient has left. The solution was developed as an Android application for tablets. Dictating was done with a headset plugged to the tablet. Support for wireless headsets was included as well. The reporting took place in a room where the nurses met patients. However, the most beneficial dictation was considered to be when nurses could dictate anywhere in the hospital they work in. Today, the situation is often that reports are typed in the end of the day based on memory and random handwritten notes. Dictating in an empty room after meeting a patient minimizes privacy issues. If dictation is to be done in rooms where patients are present, the nurses need to consider privacy issues and the patients whose information is being reported.

The speech recognition again ran on a separate server with a custom software. In the evaluations, the language the nurses were using proved to be challenging. Each special area in medicine tends to have its own terminology and accessing data to build language models is hard since privacy issues limit the use of most of the existing data. The quality of the reports is important in the medical domain. Errors in reports can lead to dangerous errors later. While the operations the participating nurses were doing and reporting were not as critical as some other operations, the system still utilized a manual proofreading step where the nurses themselves or other people in the organization with relevant rights could proofread the documents before they are archived for later use.

### 3.1.3. Crane Maintenance Reporting and Support

We developed a voice reporting application for crane maintenance specialists with Konecranes [49]. The reporting is done via an Android application. Interaction was designed to be hands-and-eyes-free with speech input and output, while speech input activation was triggered with a virtual button in a smartphone or a smart watch. As there was a definite set of content in the reporting domain, grammar-based speech recognizer was utilized. The recognizer was on a separate server, but the processing had minimal delays (a few seconds) as the network connectivity and server resources were good. Grammars

were generated dynamically by combining the target part, condition, and safety database with lists of synonyms and sentence structures. Semantic tags included in the grammars provided the semantics of the input. Each reported issue was repeated to the user with text-to-speech, along with a dialogue step to confirm the reported issue followed by related safety information and next maintenance step prompt.

The resulting report included a list of statuses on the parts inspected during the operation, with assigned next steps, so the report was partly structured. Possibility to assign free-form memos to certain inspection targets, e.g., to report what kind of lubrication was used, was considered but abandoned as it would have required open-grammar recognizer to be used. As the reports consist of fixed items, the errors are rather apparent, and users were supposed to be able to spot and correct them when checking the reports usually in the end of the day.

### 3.1.4. Elevator Maintenance Reporting

Together with KONE Corporation, we developed a voice reporting tool for elevator maintenance technicians for reporting call-out visits where there is some issue to examine and fix [50]. Besides the parsing status and issue codes, the user could dictate free-form reports. As a large vocabulary speech recognizer was required, Google's speech recognizer was chosen. Required natural language processing was handled by parsing the speech recognition results with a script using grammar definitions generated by similar script as in the crane maintenance case. As the final report model was more complex here, a state machine was used to separate the parsing of the semantic content to parts of the report, e.g., condition on arrival and the actual maintenance work. A working prototype with a graphical user interface running on a smartphone (Android) was designed and implemented. The quality of the reports in this domain was not considered to be as critical as in health care domain. The current state tends to be that reports that are typed are rather minimal and contain errors as they are usually made in the end of the day based on memory and a few handwritten notes. The users also tend to select the first somewhat suitable item from topics lists, not searching for the most suitable item. By doing the reports while working on the primary task, the goal was to improve the quality of reports as there would be more details in reports and fewer missing details and incorrectly remembered items. Voice can also make selection of the best item efficient and natural.

In both maintenance reporting cases, the conditions for which the applications were designed were expected to be noisy at times. In some cases, especially crane maintenance taking place in industrial environments, a noise-cancelling headset was considered necessary to enhance the audio quality to the level required by the speech recognition.

### 3.2. Forming the Design Space

We formed the design space in a group of four researchers who had participated in the development of the four systems described above. We started by going through each case and reminding ourselves of the design process and elements that were considered. We collected learnings and understanding gained from the development of the four prototypes and generic principles of voice interface design based on our research and giving courses related to voice interfaces over the years. We continued the process by identifying potential elements from the material. All participants raised potential aspects and they were discussed together and revised to make them suitably generic. We also identified examples illustrating each aspect and their relevance. These elements were discussed in the group until we had consensus about the meaning and relevance of each element. These elements formed the dimensions of the design space. The collection of the dimensions was done in collaborative meetings and individual researchers then wrote the descriptions of the elements to a document.

After the potential dimensions had been collected, we noticed that the resulting about 30 dimensions were too numerous to form a usable result without further processing. The group then held meetings to further arrange the design space. We organized the

dimensions into categories that we formed by discussing potential elements that could group them. We found the categories by considering widely the different elements affecting the design of voice-based reporting. We were considering the aspects affecting the designs we had done for real use contexts with various stakeholders where the work and the related organizations had significant effects. In our meetings, critical analysis of the potential dimensions and categories were agreed on. The categories were given names and their consistency and overlaps were discussed and match of each dimension considered. During the process, we adjusted these categories until consensus was reached. Once the categories had been set, we reviewed each of them once more to see if any overlapping elements were there and did final adjustments where it was considered relevant. We believe that the resulting categories form an insightful grouping of the dimensions.

## 4. Results

Our design space for voice-based professional reporting has 5 categories and 28 dimensions. It covers all the elements we identified from the four systems and their development processes. Reporting systems and voice form a complex area with many aspects. The design space presented next categorizes this area and helps designers understand and systematically tackle this design challenge. We believe the design space works as a practical tool for designers and helps them consider the various aspects that must be considered when voice-based reporting systems are designed. With the design space, these aspects can be looked at, and the needed decisions made, in a systematic manner.

### 4.1. The Design Space, Its Categories, and Dimensions

By analyzing the four voice-based reporting systems we have developed and presented in the previous section, we identified 28 elements that form the dimensions of our design space. We have further organized the dimensions into five categories: Language Processing (LP), Structure of Reporting (SR), Technical Limitations in the Work Domain (TD), Interaction Related Aspects in the Work Domain (ID), and Organization (OR). We believe these categories are aspects that can be studied separately.

Next, we introduce the 28 dimensions grouped by the five categories of the design space. For each dimension, a name, a description, possible values, and a relevant step (or steps) within the design process (see Section 4.2) are presented. The descriptions are given in a form of questions that the designers should answer. Regarding values, there are two types of dimensions: dimensions with a scale, i.e., the feature can be assessed on a scale of 0–3, and categorical dimensions, i.e., the feature has characteristics (a, b, c, . . . , n) of which one or more apply for different design cases. This information is summarized in tables. We will also explain the categories and their dimensions and demonstrate them through concrete examples from the voice-based reporting system cases presented above. The values considering each of our four systems are also provided for each dimension in the tables (Case values column), and of these, the ones discussed within the text are shown in bold.

#### 4.1.1. Language Processing (LP)

The three dimensions of the Language Processing category (Table 1) are related to language technology and the level, and type, of the processing needed or beneficial. Understanding this category is important early on so that suitable technology can be selected. If high-level processing is required, especially natural language understanding, and the language is specific to the application domain, training some dedicated models may also be needed.

**Table 1.** The dimensions of the Language Processing (LP) category deal with the requirements for transcription accuracy and the level of produced data (raw text, processed, or structured), and the potential for automatic processing. Values in bold are discussed in text.

| Dimension | Description | Values | Case Values | Steps |
|---|---|---|---|---|
| LP1: Required transcription accuracy | How accurate the results must be? Are proofreading and approval required? How harmful are recognition errors? Are there critical items? | 0: no accuracy, for own use only<br>1: basic comprehensibility<br>2: good readability<br>3: critical (proofread needed) | Lawyer: 2<br>**Nurse: 3**<br>**Crane: 1**<br>**Elevator: 1** | 4 |
| LP2: Required level of produced data | Will the recognition result be used as such or is the outcome, e.g., concepts or keywords extracted from the recognized text? | 0: raw text used<br>1: key concepts spotted<br>2: concepts and keywords spotted<br>3: full syntactic parse with semantic tags | **Lawyer: 0**<br>Nurse: 0<br>**Crane: 2**<br>**Elevator: 2** | 4 |
| LP3: Automatic processing potential | Can the results be automatically utilized, e.g., to order parts? How complex is the required natural language processing? | 0: no automation<br>1: existence of report and reporting time used<br>2: indicators automatically processed<br>3: significant operations happen automatically | **Lawyer: 0**<br>Nurse: 0<br>**Crane: 3**<br>**Elevator: 3** | 4, 7 |

For example, in the Nurse Dictation, the transcription accuracy was critical (LP1: 3) and careful proofreading and approval were required given the medical setting, i.e., treatment and dosage details. For these reasons, a medical-context-specific language model was used to make the recognition results closer to the final transcript than those that a generic language model would have resulted in. On the other hand, in the Lawyer Dictation, the requirements were less demanding as the results were used as raw material for final editing (LP2: 0) for a non-critical purpose and there was no potential, nor need, for automatic processing (LP3: 0). For the maintenance reports, basic comprehensibility (LP1: 1) was enough, although some important items needed more specific recognition: semantic information (LP2: 2) like component names and statuses, and error codes, were extracted from the recognition results. It was also envisioned that the reports would automatically support the work, e.g., relevant spare parts would be ordered automatically (LP3: 3).

### 4.1.2. Structure of Reporting (SR)

The five dimensions of the Structure of Reporting category (Table 2) concern the structure, order, schedule, and fragmentation of the reporting activity, and the possibilities to integrate the reporting to work processes. These are based on the nature of the work, how the work is organized, and, specifically, how these affect the actual reporting. To understand these issues, one must understand the work. The challenge here is that a good reporting system may also enable improving the working practices themselves, so simply following what happens currently is not a sufficient approach.

For example, in the Lawyer Dictation, the structure of the documents was free (SR1: 0) while, in the Nurse Dictation, the hospital had its own template for the structure, where certain information was expected to be entered under certain headings but these sections could contain material freely decided by the users (SR1: 2). In the Lawyer Dictation, the documents could also be dictated in any order and at any time (SR2: 0), while, in elevator maintenance, the reporting should happen when arriving on site, while working, and when leaving the site, reporting relevant information at each point (SR2: 2). In addition, there were sections "Reason for call out," "Actions," and "Components," and it was possible to navigate freely between the reporting sections. There was also a text field for free input.

In the Lawyer Dictation, the reporting happens whenever the user has available time outside the main work (SR3: 0), i.e., meeting with a client, and the reporting can be fragmented over the workday interleaved with other work (SR4: 3). In both maintenance cases, the reporting can be part of the work (SR3: 3) and it can happen whenever something is done (SR4: 2), even during a task (SR5: a). In the Nurse Dictation, however, the report is made after the patient has left (SR4: 1) and the reporting can happen (before or) after the main task (SR5: b).

**Table 2.** The dimensions of the Structure of Reporting (SR) category are related to structural characteristics of the final report and how the structuring can, should, or must be done. Values in bold are discussed in text.

| Dimension | Description | Values | Case Values | Steps |
|---|---|---|---|---|
| SR1: Structure of the report | How structured must or should the final document be—fully structured, semi-structured, or unstructured? | 0: no structure<br>1: top level headings<br>2: sub level headings<br>3: each information item has dedicated section, fully structured document | **Lawyer: 0**<br>**Nurse: 2**<br>Crane: 3<br>Elevator: 3 | 1 |
| SR2: Reporting order | How fixed is the order of data entry? | 0: no limitations<br>1: main segments in specified order<br>2: subheadings in specified order<br>3: order fully fixed | **Lawyer: 0**<br>Nurse: 0<br>Crane: 3<br>**Elevator: 2** | 2 |
| SR3: Reporting and main task | Can reporting be integrated to work processes? | 0: no integration, reporting separately<br>1: reporting done between main tasks<br>2: reporting done between different subtasks<br>3: reporting part of the work | **Lawyer: 0**<br>Nurse: 1<br>**Crane: 3**<br>**Elevator: 3** | 2 |
| SR4: Fragmentation of reporting | Does reporting happen in one, uninterrupted session or in fragments? | 0: in a single session (e.g., end of day)<br>1: after each main task<br>2: after subtasks<br>3: reporting continuously as part of primary task | **Lawyer: 3**<br>**Nurse: 1**<br>Crane: 2<br>Elevator: 2 | 2 |
| SR5: Schedule of reporting | What are suitable times to do the reporting, e.g., in the field? | A: reporting during the primary task<br>b: reporting before and after the task<br>c: reporting while in transit<br>d: reporting in the end of the day | Lawyer: b, c, d<br>**Nurse: b**<br>**Crane: a**<br>**Elevator: a** | 2, 7 |

### 4.1.3. Technical Limitations in the Work Domain (TD)

The seven dimensions of the Technical Limitations in the Work Domain category (Table 3) deal with privacy and security issues, possible and acceptable delays, and the requirements set by domain-specific laws or language, e.g., the role of the report within the work process. The domain of the work not only affects the reporting structure but sets many practical limitations and demands as well. Technical limitations are aspects that set requirements and constraints to the software and, to an extent, to hardware. In addition, the category contains dimensions related to the structure and content of the final document, to the extent defined by the generic application domain, rather than an individual organization and its rules.

Different elements of audio processing may be placed in a local device, into cloud, or, in some cases, the processing can be done near the terminal device. Examples of this include audio processing required for visualization, which, in the Lawyer Dictation, was done using local processing to make recording and editing responsive (TD1: 0). Transmitting the data can vary depending on the network conditions, which may vary greatly for field workers visiting different sites.

Actual speech recognition can require significant processing power, and, in the Nurse Dictation, the processing could not be done on public cloud and caused significant delays (TD2: 1). In maintenance reporting cases similarly server-/cloud-based processing had short delays (TD2: 2). The needs of these applications made the delays inconvenient at times, but not unacceptable. The acceptable delays from full voice processing varies in the example cases from an acceptable 24-h delay in the Nurse Dictation (TD2: 1) to maintenance reporting where real time processing (TD2: 3) is beneficial.

**Table 3.** The dimensions of the Technical Limitations in the Work Domain (TD) category are about the limitations that the domain puts to speaking out loud the reported information and the related speech processing. Values in bold are discussed in text.

| Dimension | Description | Values | Case Values | Steps |
|---|---|---|---|---|
| TD1: Processing delays | What kind of delays the processing and data transmission may create? | 0: local processing, minimal delays<br>1: distributed processing, delays on results<br>2: cloud/server processing, short network delays<br>3: cloud/server processing, long network delays | **Lawyer: 0**<br>Nurse: 3<br>Crane: 2<br>Elevator: 2 | 4, 6 |
| TD2: Real-time requirements | What are the processing time requirements and acceptable delays? How soon will the results be valuable for different parts of the organization? | 0: no requirements (e.g., one-week is acceptable)<br>1: report must be ready and proofed within 24 h<br>2: report must be ready within 1 h<br>3: report must be ready with one minute | Lawyer: 2<br>**Nurse: 1**<br>**Crane: 3**<br>**Elevator: 2** | 4, 7 |
| TD3: Privacy and Information Security | Can data transfer and processing cause potential privacy or security issues? Should the speaker remain anonymous? | 0: no security concerns<br>1: encrypted data transfer<br>2: secured device identity and transfer<br>3: data transfer via secure network | **Lawyer: 1**<br>**Nurse: 3**<br>Crane: 2<br>Elevator: 2 | 5 |
| TD4: Dictation privacy and security per locations | Is there risk of privacy or security issues due to reporting location and public nature of speech? (eavesdropping) | 0: no risks (reporting in company facilities)<br>1: limited risk, non-public spaces<br>2: in public, non-critical information<br>3: critical information, unsecure locations | Lawyer: 2<br>**Nurse: 0**<br>**Crane: 1**<br>**Elevator: 1** | 5 |
| TD5: Formal requirements of the domain | Are there legal and comparable requirements imposed to the documents? Are there traditions in the field to follow? | 0: no requirements<br>1: reporting required<br>2: content of the reports specified<br>3: reporting practices, format, and usage regulated | **Lawyer: 0**<br>**Nurse: 3**<br>Crane: 2<br>Elevator: 2 | 1 |
| TD6: Language of the Domain | How precise and formal is the language in the domain? How much is there domain specific terminology and jargon? | 0: no specifics<br>1: domain jargon<br>2: organization has own terminology<br>3: specific language must be used (e.g., legislative restrictions) | **Lawyer: 3**<br>**Nurse: 2**<br>**Crane: 1**<br>Elevator: 1-2 | 4 |
| TD7: Role of the final document | Is the document the main result of the work, part of process, or does it support work and improve efficiency? | a: report is the primary result<br>b: report is part of the result<br>c: report is supportive or for quality control<br>d: report for personal use (note taking vs. reporting) | **Lawyer: a**<br>**Nurse: b**<br>**Crane: c**<br>**Elevator: c** | 7 |

The cases also illustrate different data security requirements. In the Nurse Dictation, the health-related audio and reports had to reside on a secure server and be transmitted over a secure connection. Clients had also security tokens installed (TD3: 3). In the Lawyer Dictation, encryption of data transfer (TD3: 1) was enough.

Dictation location varied in the example cases as well. In the Nurse Dictation, the location was a room in the hospital with no access by external people (TD4: 0), while maintenance reporting happens in customer premises where not only the customer's employees but also outsiders may be present. Furthermore, communications towards customers required more formal and careful reporting and some information is relevant to the reporting side only. For example, in maintenance some technical details could confuse customers (TD4: 1) and, therefore, the users must consider what and where they report.

External requirements for the resulting document vary in the examples from the Nurse Dictation, where the reports must fulfill the requirements of the medical field (TD5: 3), to the Lawyer Dictation, where there were no external requirements (TD5: 0) and each

lawyer uses their expertise throughout the process. Similarly, the language used varies while each of the examples has domain-specific words, phrases, and expressions. In the Nurse Dictation, the language was very specific to the type of nursing the users do (TD6: 2). In maintenance reporting, jargon and terminology of the domain and the companies is present (TD6: 1-2). In the Lawyer Dictation, there was a significant amount of words not used in average communication (TD6: 3).

Finally, the role of the resulting document is important, and in some domains, a single report plays multiple roles. The examples include the Lawyer Dictation where the report is the primary result (TD7: a), the Nurse Dictation where it supports future work with the patient (TD7: b), and in maintenance reporting it helps the organization (TD7: c).

### 4.1.4. Interaction Related Aspects in the Work Domain (ID)

The seven dimensions of the Interaction Related Aspects in the Work Domain category (Table 4) consider aspects that the domain sets to the process on how the reports can be created in practice, i.e., what kind of limitations to the human-technology interaction the domain sets. Many of these limit what the user can do but some are more requirements for the applied technology.

**Table 4.** The dimensions of the Interaction Related Aspects in the Work Domain (ID) category limit the design because of the physical load and limitations imposed by the actual work and environment where the work is done. Values in bold are discussed in text.

| Dimension | Description | Values | Case Values | Steps |
|---|---|---|---|---|
| ID1: Physical load | Is the user under physical load while reporting? | 0: no physical load<br>1: light physical load<br>2: moderate physical load, limits speech<br>3: heavy physical load, speech impossible during task | **Lawyer: 0–1**<br>**Nurse: 0**<br>**Crane: 3**<br>Elevator: 2 | 6 |
| ID2: Cognitive load | Is there cognitive load to the user due to main/other tasks while reporting? | 0: no other cognitive load<br>1: limited cognitive load<br>2: significant cognitive load, limiting reporting at the same time<br>3: severe cognitive load, reporting at the same time not possible | Lawyer: 0–1<br>**Nurse: 0**<br>**Crane: 1–3**<br>**Elevator: 1–3** | 3 |
| ID3: Reserved resources | Are user's hands and/or eyes busy while reporting? | 0: hands and eyes can be used freely<br>1: hands and/or eyes can be used most of the time<br>2: hands/eyes busy most of the time<br>3: hands/eyes busy all the time | Lawyer: 0–2<br>**Nurse: 0–2**<br>**Crane: 2**<br>**Elevator: 2** | 3, 6 |
| ID4: Noise | Are the reporting conditions noisy? | 0: no noise (quiet office)<br>1: low background noise<br>2: varying noise, at time prevents reporting<br>3: extreme noise, speech-based reporting impossible | Lawyer: 0–1<br>**Nurse: 0–1**<br>**Crane: 3**<br>Elevator: 2 | 3, 4, 6 |
| ID5: Devices and Speech Technology | What kind of devices are and can be used? What is the potential for local or remote processing and what are the related delays? | a: local hardware for all processing<br>b: preprocessing in local hardware, server-based recognition and language processing<br>c: server-based processing | **Lawyer: c**<br>**Nurse: c**<br>**Crane: b, c**<br>**Elevator: b, c** | 4, 6 |
| ID6: Interaction modalities | What interaction modalities (input, output) can be applied to the reporting interface? | a: speech input<br>b: gesture input<br>c: buttons (e.g., in work clothing)<br>d: audio output<br>e: visual output, small display (smart watch)<br>f: visual, large display (mobile device)<br>g: haptic feedback | Lawyer: a, c, f<br>**Nurse: a, c, d, f; a, e**<br>**Crane: a, c, d, e, g**<br>**Elevator: a, b, c, d, e/f, g** | 3<, 6 |
| ID7: Speech activation method | What are the suitable speech input activation methods–buttons, touch screens, or audio processing? | a: push to talk<br>b: press to talk, press to stop<br>c: press to talk, VAD detects end<br>d: keyword for start, VAD for end<br>e: automatically detected start and end | **Lawyer: a**<br>Nurse: b<br>**Crane: c**<br>**Elevator: c, d** | 6 |

For example, in the Nurse Dictation the users had no physical load while dictating (ID1: 0); in Lawyer Dictation, the user may be traveling while dictating (ID: 0–1); and

in maintenance work, there can be, at times, heavy load preventing reporting (ID1: 3). Similarly, in the Nurse Dictation, there was no other cognitive load while reporting (ID2: 0), while in maintenance work, the users may be under severe cognitive load at times (1–3).

In the Nurse Dictation, users' hands and eyes were free if they were reporting after a meeting with a patient but while moving in a ward hands may be reserved (ID3: 0–2), and in maintenance reporting, both hands and eyes are often reserved (ID3: 2).

In the Nurse Dictation, reporting conditions are quiet if the reporting takes place in an empty room but in wards, there can be occasional noise (ID4: 0–1). In crane maintenance reporting, the conditions can be noisy at times (ID4: 3).

In all cases, server-based processing was acceptable (ID5: c). However, in some cases maintenance work may be in locations with limited network connectivity (ID5: b).

In the Nurse Dictation, all modalities as part of a tablet-based interface were available (ID6: a, c, d, f) when reporting in an empty room. In wards, the set can be more limited (e.g., ID6: a, e). In maintenance reporting, busy hands and eyes and physical work limit access to buttons (ID6: c, a single button possible); haptic feedback (ID6: g) may be hard to feel, and display must be easy to reach (smart watch or similar). Simple gestures may be possible (ID6: b). Speech activation was similarly different: in the Lawyer Dictation, push to talk was suitable (ID7: a), but in maintenance reporting, while using a button or a keyword to start the reporting is possible (when conditions are not overly noisy), ending it should be based on voice activity detection (VAD) (ID7: c, d).

### 4.1.5. Organization (OR)

The six dimensions of the Organization category (Table 5) deal with requirements that the organization and its practices set to reporting systems. These aspects capture the practical and formal arrangements of the organization where the reporting takes place. While the organization can also change when improvements to reporting are done, many of the aspects cannot be flexible as reporting is linked to other processes and information systems. The users are also part of this category as they are employees of the organization and receive training as a part of their work.

For example, in the Lawyer Dictation, the users decided how they wanted to work (OR1: 0), while in maintenance reporting certain protocol must be followed when visiting customers and fixed things should be reported (OR1: 3).

How reporting is connected to other systems varied from the Lawyer Dictation, where emailing the resulting documents was sufficient (OR2: 0), to the Nurse Dictation, where the report needs to be transferred to an existing patient information system (OR2: 2). When designing maintenance reporting, it become evident that the information systems can provide information that can both direct the reporting and support the work (OR2: 3).

While by definition in professional reporting all users have some expertise, the level and type of both education and training can vary greatly. In the Nurse Dictation, all users were trained nurses (OR3: 3). In maintenance reporting, all users have certain level education but there can be significant variation (OR3: 2).

In all of our example cases, users may have different native tongues, but all must be able to speak the language of the organization (OR4: 2). Maintenance organizations are often international, and branches have different types of workers (OR4: 0). Furthermore, in international organizations, local branches often have different operating languages.

In the Lawyer Dictation, the original user was the user of the document (OR5: a). In maintenance reporting, in addition to archiving the maintenance history of individual equipment, the report could be used for quality control, development of predictive data analytics, logistics support, and so on (OR5: f), and different users of a report may have very different needs. In the Lawyer Dictation, there was only one person contributing to the report (OR6: a). In the Nurse Dictation, in turn, another person proofread the document (OR6: c).

**Table 5.** The dimensions of the Organization (OR) category guide designers to understanding the rules of the organization that will use the reports. Values in bold are discussed in text.

| Dimension | Description | Values | Case Values | Steps |
|---|---|---|---|---|
| OR1:<br>Rules and practices of the organization | Are there strict rules to follow or flexible practices regarding reporting? | 0: no rules<br>1: generic instructions on what and how to report<br>2: overall reporting process specified<br>3: strict rules for process, content and structure | **Lawyer: 0**<br>Nurse: 2<br>**Crane: 3**<br>**Elevator: 3** | 1 |
| OR2:<br>Backend system | How is reporting linked to other information systems? How much these systems provide information which affects the speech interface? | 0: no links<br>1: report stored in organization level system with other materials<br>2: report goes to systems in structured format<br>3: other systems and external information, possibly from multiple organizations, directs reporting | **Lawyer: 0**<br>**Nurse: 2**<br>**Crane: 3**<br>**Elevator: 3** | 1, 7 |
| OR3:<br>Users | What are the backgrounds of users and what is their education and training level? | 0: all kinds of users<br>1: all users have certain education<br>2: users have certain education and training<br>3: users have matching education and training | Lawyer: 1<br>**Nurse: 3**<br>**Crane: 2**<br>**Elevator: 2** | 3 |
| OR4:<br>Multilingual. multicultural users | Is the user population multilingual and/or multicultural? | 0: users from different cultures<br>1: all can speak the same language (not necessarily natively)<br>2: proficient in organization's language<br>3: matching background and language fluency | **Lawyer: 2**<br>**Nurse: 2**<br>**Crane: 0, 2**<br>**Elevator: 0, 2** | 5 |
| OR5:<br>Users of the document | Are there different uses and users for the document aside from the original user? | a: reporter only (personal note taking)<br>b: peers<br>c: multiple people in the organization<br>d: used across organizations<br>e: used to plan future<br>f: multiple purposes | **Lawyer: a**<br>Nurse: b, c<br>**Crane: f**<br>**Elevator: f** | 5, 7 |
| OR6:<br>Actors contributing to the report | Do multiple people contribute to the document (includes proofreading)? | a: one person per report<br>b: report is checked and confirmed by others<br>c: proofread and edited by another person<br>d: multiple people contribute<br>e: document which evolves over long time. | **Lawyer: a**<br>**Nurse: c**<br>Crane: a/e<br>Elevator: a/e | 5, 7 |

### 4.2. The Design Process of Voice-Based Reporting Systems

The proposed design space consists of a large collection of dimensions across the five categories. Understanding all of these is important for a designer as misunderstanding with just one aspect can result in a solution that is not suitable for the users or the organization. To help organize the design work, we suggest the sequence of seven steps to be taken in the order presented in Figure 1. The process directs design to systematically identify important information the beginning. The design work related steps are (1) "Report": understanding the report to be produced, (2) "Reporting": understanding the reporting activity and what is possible and important, (3) "Interaction": understanding what kind of interaction solutions can be utilized, (4) "Language technology": understanding current potential of language technology, (5) "Privacy and security": understanding privacy and security issues involved, (6) "Devices": getting aware of devices that could be used, and (7) "Report processing": understand how the report will be processed and used after it has been done. In order to gather the required understanding at each step and finish with a successful professional reporting VUI design, designers need to utilize different information sources and methodologies.

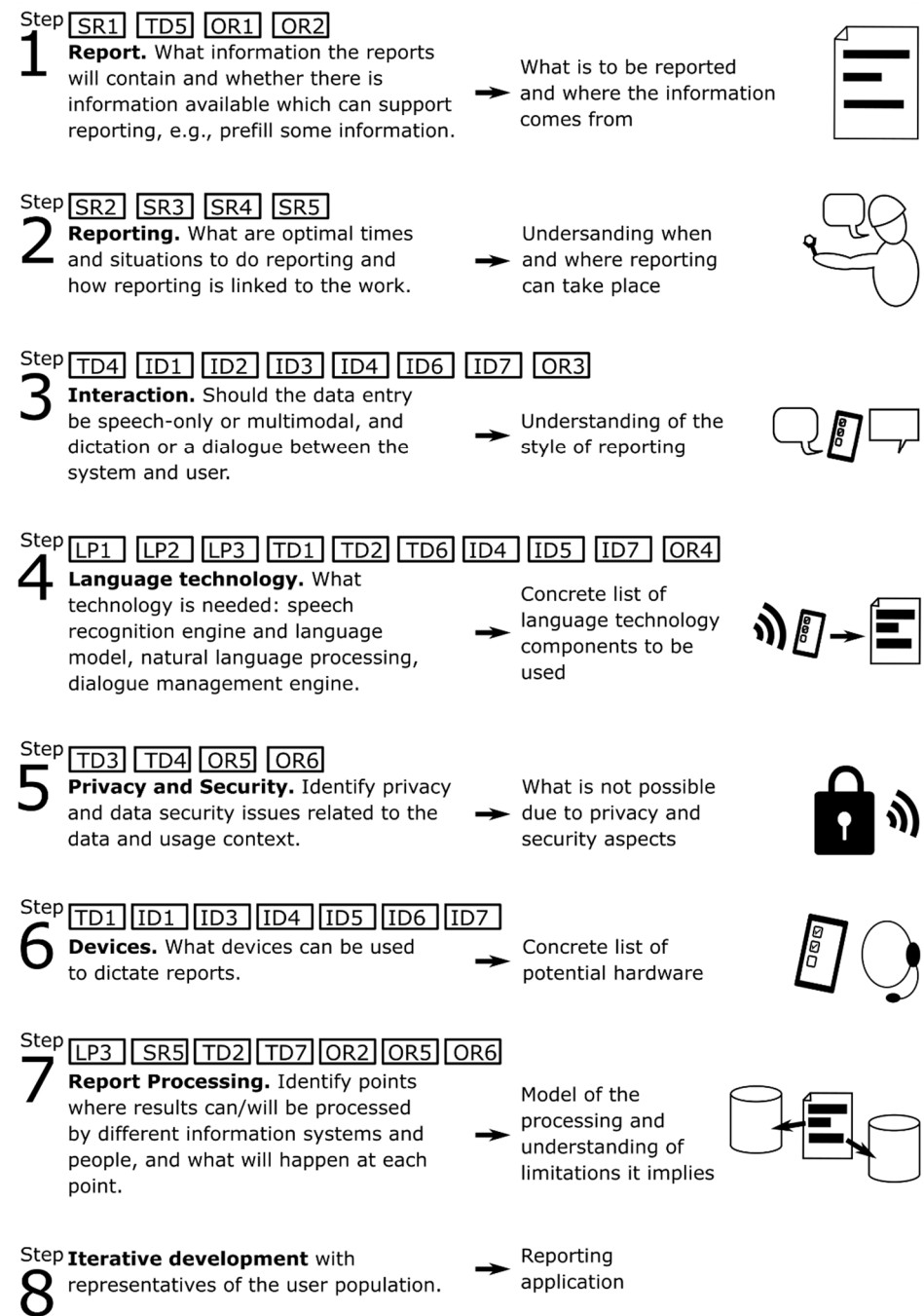

**Figure 1.** The design process of voice-based reporting systems consisting of eight steps. The dimensions of the design space that must be understood at each step are listed inside rectangles and the outcome of each step summarized on the right.

Throughout the steps various people must be involved. Steps 1 "Report," 5 "Privacy and security," and 7 "Report processing" require input from people representing the organization where the reports will be created and used. IT departments are in most cases where the understanding of current and future systems is available. Information security is also understood by the same group of people. Reports also have many uses in organizations. As an example, in elevator maintenance, call-out reports are used to understand the root causes of the failure and to improve predictive data analytics. Therefore, data analysts must have a say on what information should be gathered and how it should be organized.

Management level of the target organization is often the best source in Step 1 "Report" and Step 7 "Report processing." In addition to eliciting information from individuals, access to relevant documentation can be valuable. In many fields, the concepts used in reporting are specified in handbooks, many kinds of technical documents, and databases. Lists of various kinds can be utilized, e.g., lists of part numbers, error codes, or process steps. However, the language used in written documents is not always directly usable in voice-based reporting. Designers should identify how different things are referred to in spoken communication. Especially complex codes can be unnatural in spoken form. Still, the formal documents can be a valuable starting point.

Representatives of the end user population can help especially in Step 2 "Reporting" as their current daily work and their needs and wishes must be understood. End users can provide information about their current practices, and with suitable methods, one can also gather understanding on what kind of reporting-related work practices would be accepted in the future. Producing prototypes of various fidelities, from video mockups to low-fidelity interactive prototypes, is valuable in this work. Ethnographical approaches of observing the current work can also provide insights.

Identifying suitable technologies in Step 4 "Language technology" often happens in co-operation with experts of a language technology supplier company. This is also linked to Step 6 "Devices." With a good understanding of the language and structure of the dialog, and the acoustic conditions where the reporting is to happen, the communication with the technology provider(s) can be efficient and right solutions identified early on. The privacy and information security demands must also be communicated to the technology provider.

## 5. Discussion and Conclusions

Speech technology has reached the level where voice-based reporting can be used in various professional areas. Using voice for reporting can be an efficient way to generate reports and voice-based reporting may also result in higher quality reports than other methods. With well-designed solutions, the reporting may also become part of the work. This can improve work efficiency, but more importantly, it can improve the quality of reporting as the reports can be done in real time without the need to neither remember details later nor make, or interpret, handwritten notes.

This approach can be expanded also beyond professional reporting. As an example, we are looking at voice-based input to collect feedback from the people in built environments, such as office environments. The organizations maintaining buildings value feedback from users as it enables them to react to issues in a timelier manner. Logging in to an intranet to fill in a feedback form is often too big of a step to report minor issues. By asking for voice-based feedback in relevant spaces at relevant times, the amount and relevance of feedback can be increased.

However, developing good voice-based reporting solutions requires understanding of the work and the context beyond what is usually required in user centered processes. Most importantly, since voice-based reporting systems deal with language, understanding what kind of language will be used in the reports is vital. Potential speech technology must then be tested with examples of this language to see if it must be tuned to support the language properly. One should also carefully analyze the work and the context where it is performed. The public nature of speech must be considered together with the context of reporting. As others can hear what is being spoken, unless the reporting is done in closed room or in a car, privacy and information security must be considered.

We have presented a design space for voice-based professional reporting systems. It emphasizes the multitude of aspects that designers need to consider when voice-based reporting solutions are to be created. Voice-based reporting, when well developed and integrated to work, can improve both the quality and efficiency of reporting by making reporting a natural and pleasant part of the work. At best, the reporting can benefit greatly from the hands-and-eyes-free potential of voice input and the power of natural language.

When reporting can be integrated to the work, the results can be pleasant for the user and a good reporting system can support the primary task.

The design space and the related process we have presented can help designers in building successful voice-based reporting solutions. Technology is quite ready for wide application of speech technology in many work domains, so we believe these issues are increasingly relevant to many designers.

**Author Contributions:** Investigation, J.H., T.K., M.T., and S.S.; methodology, J.H., T.K., and M.T.; supervision, M.T.; visualization, J.H. and T.K.; writing–original draft, J.H., T.K., M.T., and S.S. All authors have read and agreed to the published version of the manuscript.

**Funding:** The reporting systems discussed in this article were developed in various projects. The related research projects and their funders are as follows: Lawyer Dictation—project *Mobile Voice Editing and Speech Recognition Software* (MOBSTER) was funded by the industry; Nurse Dictation—project *Mobile and Ubiquitous Dictation and Communication Application for Medical Purposes* (MOBSTER), Crane Maintenance Reporting and Support—project *Smart technologies for lifecycle performance* (S-STEP), and Elevator Maintenance Reporting—project *Dynamic visualization in Product/Service Lifecycle* (DYNAVIS) were funded by the former Tekes—the Finnish Funding Agency for Technology and Innovation, now part of Business Finland.

**Acknowledgments:** We thank Juho Hella for his contributions to the process of building this design space and to many of the described reporting solutions. We also thank our collaborators in Mobiter Dicta, University of Turku and Turku University Hospital, Konecranes, and KONE Corporation.

**Conflicts of Interest:** The authors declare no conflict of interest. The funders had no role in the design of the study; in the collection, analyses, or interpretation of data; in the writing of the manuscript, or in the decision to publish the results.

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
