# Peer review of "Design Space for Voice-Based Professional Reporting"

_mti, doi:10.3390/mti5010003_

Round 1

Reviewer 1 Report

This paper presents a design space for voice-based reporting systems based on the research team’s prior experience in designing and evaluating four such systems in different professional contexts. The design space takes the form of 28 “dimensions”, each of which is defined by a question about the system and several possible answers. These dimensions are grouped into five categories, relating to different aspects of the reporting task.

This is a practical paper that seems as though it would be useful to anyone designing a voice-based reporting system. The rationale and motivation for the paper is quite strong; voice is a rather complex area of design with many pitfalls, which many designers are working in for the first time at the moment. While many papers evaluate one VUI system, it is good to see some broad lessons drawn out from multiple iterations of designing a similar system (i.e. voice reporting) for different contexts. To my knowledge, this is a novel contribution.

The reviews of prior literature on voice-based reporting and design spaces are clear, and the four VUI development projects are explained in sufficient detail for the purposes of this paper.

Overall, I rate this paper as a solid contribution to the literature. I have a few points to critique that mainly concern the clarity of its reporting. These should not be too difficult to address.

The description of how the dimensions were developed (section 3.2) is vague. I would like to see a few sentences describing how the researchers approached the task of coming up with the initial design space considerations. Currently, the paper describes this as “the collection of potential elements recognized from the material”. This is very passive and indicates that the considerations arose on their own rather than being created by the researchers. It would be more appropriate to acknowledge that the considerations were actively developed by the researchers based on their reflections on their own experiences.

Section 4.1 (second paragraph) discusses five “elements” that must be considered, but the point of this discussion was not entirely clear. From the main text, it seems like those five elements were considered but ultimately dropped in favour of the final set of categories; however, figure 1 implies that the five elements are part of the contribution of the paper (or at least that was my impression). I also could not draw a clear understanding of what figure 1 is trying to communicate, partly because the placement of the bubbles is not intuitive to me. E.g. why is “users” part of Technical Domain but “software” is not? Why is “hardware” part of Organisation but “work” is not? It seems to me that this section may be clearer if figure 1 is simply removed, but I may be missing something about the point that it is supposed to make.

The core results of the paper are well presented overall. However, there is some inconsistency in what the dimensions are asking about. Some dimensions seem to be asking for a description of the actual capabilities of an existing system, while others seem to be asking for a description of the acceptable standards that a system has to meet. The wording of the dimensions could be improved to make this more consistent. (The paper could also note that the schema might be used for either purpose, as a side-by-side rating of the requirements and actual capabilities of a system side-by-side would effectively constitute a heuristic evaluation checklist.)

Much of the core content of the paper is embedded in the tables and figure 2, without a full description of each point in the body text. That makes it crucial that those parts of the paper are clear, but in some places the wording is squashed to the point of obscurity. (E.g. “matching background and language fluency” OR4 is a sentence fragment that I had to read several times to understand.) I suggest a light revision of the wording of the tables as well as the text in figure 2 to ensure that each point is clear and unambiguous. Maintaining a consistent format would help – e.g. in figure 2, some of the steps describe an action while others describe an outcome or pose a question.

In section 4.2, it would be helpful for the body text to refer to the eight steps by name rather than number, and for those names to be immediately understandable; e.g. “Step 1 (Designing the Report)". I would also suggest including the stakeholder groups that are described in section 4.2 on figure 2, although that is merely a suggestion.

My final point in relation to clarity is that the paper uses the passive voice extensively, and that makes it hard to follow in some places. The paper would be easier to read if it favoured the active voice more often. This is not a requirement for acceptance, but a point of feedback for future papers.

Author Response

Thank you for your valuable review.

We have edited section 3.2 to more clearly describe the process of how the design space was formed adding a few sentences and revising some expressions which we believed we could phrase with more concrete words. (Lines 269-296 in the revised manuscript)

Based on your comment, we reconsidered Figure 1. We have had discussions about its value and if it is worth including it already before and after the review commend and a round of discussion decided to remove it. The text segment related to the figure was similarly removed from Section 4.1. We do believe cutting this extra branch improves the paper. (Lines 301-318 in the original manuscript, including Figure 1 removed. Figure number in to refer to the remaining Figure has naturally been updated in line 495 of the revised manuscript)

In the tables 2.-5, descriptions of the dimensions (in column "Description") were revised to make them all have the same form.

Section 4.2 now refers to process phases with their names, not just numbers. (Lines 493-595 in the revised manuscript)

The text has also been edited to replace unnecessary passive voice with more explicit expressions. (Lines 73, 180, 181, 283, 285, 287, 292, 293, 295 in revised manuscript)

Minor language editing has been done across the text (at least lines 45, 178-179, 212, 213, 434-435, 

Reviewer 2 Report

The manuscript is very well-written and was pleasing to read. The content is well-organized. The choice of components to be included in the design space is explained in detail. Also, the presentation of methodology of designing is satisfactory. Visualizations (tabular) are very good and self-explanatory. 

The only prominent suggestion that I have is, the effectiveness of the presented design space is very intuitive. It would be great if the evidence for this effectiveness is presented quantitatively. You may include some measures like recognition efficiency or word error rate of speech recognition system, with and without, all or some of the components of the design space. This will significantly improve the scientific significance of the article.

Minor comments: 

 1) Page 2: line no: 45 - Speech technologies themselves are not familiar "with" too many designers ...

 2) Page 2: line no: 75 - 'This paper closes with ...'. Please rewrite.

 3) Page 8: line no: 331 - "re related". Please correct !

 4) Page 11: Table 3: Row 2: What kind of delays processing and data transmit may create? -> What kind of delays that the processing and data transmission may create?

Author Response

Thank you for your review.

We have edited the parts of the manuscript that contained the errors your identified in your review. In the revised manuscript edits related to your comments can be found in lines 45, 76-77, 371 and first row of table 3, column titled "Description".

Most of the tables have received minor editing to "Description" column to make the dimension description more uniform in style.

Regarding your suggestion to have quantitative data to support the effects of the design dimensions, such analysis would indeed be interesting and create a strong contribution. However, with the current data we are not able to provide such comparisons as this would need to be considered throughout the research work. Perhaps we will have such an opportunity in the future.

Other edits that can be seen in the revised manuscript are replacing passive voice with active one to remove ambiguity, removal of Figure 1 and it has been considered unnecessary distraction and minor language editing across the text.